# Occurrence of Extended Spectrum Beta Lactamase (ESBL) Producers, Quinolone and Carbapenem Resistant *Enterobacteriaceae* Isolated from Environmental Samples along Msimbazi River Basin Ecosystem in Tanzania

**DOI:** 10.3390/ijerph18168264

**Published:** 2021-08-04

**Authors:** Zuhura I. Kimera, Fauster X. Mgaya, Stephen E. Mshana, Esron D. Karimuribo, Mecky I. N. Matee

**Affiliations:** 1Department of Microbiology and Immunology, Muhimbili University of Health and Allied Sciences, P.O. Box 65001, Dar es Salaam 11103, Tanzania; fauster.mgaya@sacids.org (F.X.M.); mateemecky@yahoo.com (M.I.N.M.); 2Ministry of Livestock and Fisheries, Mtumba Area, P.O. Box 2182, Dodoma 40487, Tanzania; 3Department of Microbiology and Immunology, Catholic University of Health and Allied Sciences, P.O. Box 1464, Mwanza 33109, Tanzania; stephen72mshana@gmail.com; 4SACIDS Africa Centre of Excellence for Infectious Diseases, Sokoine University of Agriculture, P.O. Box 3297, Morogoro 67125, Tanzania; karimuribo@sua.ac.tz; 5Department of Veterinary Medicine and Public Health, Sokoine University of Agriculture, P.O. Box 3021, Morogoro 67125, Tanzania

**Keywords:** effluent, river water, river sediment, crop soil, antimicrobial resistance, Msimbazi river basin, *Enterobacteriaceae*

## Abstract

We conducted environmental surveillance of antimicrobial resistance (AMR) bacteria in the Msimbazi river basin in Tanzania to determine the occurrence of extended-spectrum β-lactamase (ESBL)-producing, carbapenem resistant *Enterobacteriaceae* (CRE) and quinolone resistant *Escherichia coli* and *Klebsiella* spp. A total of 213 *Enterobacteriaceae* isolates were recovered from 219 samples. Out of the recovered isolates, 45.5% (*n* = 97) were *Klebsiella pneumoniae* and 29.6% (*n* = 63) were *Escherichia coli*. *K. pneumoniae* isolates were more resistant in effluent (27.9%) compared to the *E. coli* (26.6%). The *E. coli* had a higher resistance in river water, sediment and crop soil than the *K. pneumoniae* (35 versus 25%), respectively. Higher resistance in *K. pneumoniae* was found in nalidixic acid (54.6%) and ciprofloxacin (33.3%) while the *E. coli* isolates were highly resistant to ciprofloxacin (39.7%) and trimethoprim/sulfamethoxazole (38%). Resistance increased from 28.3% in Kisarawe, where the river originates, to 59.9% in Jangwani (the middle section) and 66.7% in Upanga West, where the river enters the Indian Ocean. Out of 160 *E. coli* and *K. pneumoniae* isolates, 53.2% (*n* = 85) were resistant to more than three classes of the antibiotic tested, occurrence being higher among ESBL producers, quinolone resistant and carbapenem resistant strains. There is an urgent need to curb environmental contamination with antimicrobial agents in the Msimbazi Basin.

## 1. Introduction 

Antimicrobial resistance (AMR) presents a major threat to public and animal health, the global economy and security [1]. The World Health Organisation considers AMR to be one of the top ten threats to global health [2]. Without interventions, by 2050 at least 50 million people will die annually [3]. Due to the nature and complexity of the drivers of AMR, efforts to control AMR must utilise a One Health based approach [1,4,5]. Although tackling AMR requires surveillance of human–animal environment compartments, the latter is relatively under-investigated, especially in sub-Saharan African countries [6]. This is largely due to a lack of proper understanding of the role of the environment as a conduit and reservoir of AMR pathogens and genes, with potential spillover to animals and humans [7]. Our previous study indicated spillage of resistant environmental isolates into Lake Victoria through the sewage system in the city of Mwanza, involving clonal spread of resistant strains as well as dissemination by commonly occurring IncY plasmids [8]. 

Although the presence of AMR micro-organisms in the environment is of global concern, the situation in Africa is more critical due to weak regulation in the use of antimicrobials, weak surveillance systems for antimicrobial use (AMU) and AMR, unregulated disposal of waste and poor sanitation infrastructure [9,10,11,12]. This is coupled with a lack of basic knowledge of the concept of antimicrobial resistance [13,14] and a lack of continuing medical/veterinary education for prescribers [15]. In addition, the use of antimicrobials as pesticides in crops is very common [16]. 

Yet in our review of AMR studies conducted in Africa between 2005 and 2018, we found only ten (5.68%) out of 176 published investigations on AMR involved the environment [6]. Four of these studies were conducted in South Africa, two in Algeria and one in each of the following countries: Ethiopia, Egypt, Tunisia and Tanzania. These studies, which sampled domestic and biomedical waste, wastewater, river sediments, surface and drinking water, treated wastewater, river water and vegetables, found the prevalence of MDR *E. coli* ranging from 33.3 to 100%, with resistance to 16 different antimicrobial agents. The variations in the findings could be attributed, at least in part, to methodological differences [6]. 

Recently, a framework for environmental surveillance of antibiotics and antibiotic resistance has been developed, with a choice of markers and sampling sites for harmonizing surveillance systems under different scenarios [7]. For each scenario, the framework provides suggestions on different phenotypic and genotypic microbial surveillance markers, as well as antibiotic residues and sites where monitoring would be particularly informative. Such an approach ensures good quality data that can be shared between countries and contribute toward the protection of human, animal and ecosystem health [17]. 

Using this framework, we conducted environmental surveillance of AMR bacteria in the Msimbazi river basin in Dar es Salaam, Tanzania’s largest city and one of Africa’s fastest growing metropolitan areas [18]. The city’s growth is largely centred along the Msimbazi Basin, with 27% of its population living in the basin and along its tributaries, which flow through the heart of Dar es Salaam [19]. The basin, which is important to the city’s development, environment and economy, has many human activities of significance with respect to the emergence and spread of AMR in the environment. Unfortunately, service provision in the basin has not kept up with the rate of urbanisation, resulting in unplanned settlements, sanitation challenges, pollution, inadequate infrastructure and erosion. Our focus was on the occurrence and distribution of extended-spectrum β-lactamase (ESBL)-producing and carbapenem resistant *Enterobacteriaceae* (CRE) and quinolone resistant *Escherichia coli and Klebsiella* spp. in environmental samples. Many enterobacteria that have been isolated from environments, such as *E. coli and Klebsiella* spp., are often associated with ESBL production and carbapenem and quinolone resistance, often occurring together in most multidrug resistant phenotypes [20,21,22]. Resistance to these classes of antibiotics is very significant, since it renders most of the drugs used for human and veterinary medicine ineffective [6,14]. 

## 2. Materials and Methods

### 2.1. Study Area

This study was conducted between February and March 2021 in the Msimbazi River basin in Tanzania. The river originates from the Kisarawe highlands in the Pwani Region and discharges its water into the Indian Ocean as previously described [23]. The Msimbazi River serves as an important drainage system for rainwater, runoff water and wastewater from community, medical, industrial and animal farming and agricultural activities along the basin.

### 2.2. Sampling Frame

Using integrated surveillance for the collection of AMR data in the environment as described in [7], a sampling frame from the study area was generated. The specified sampling frame for the study included poultry and domestic pig farms, crops and vegetable farms that use manure from the domesticated animals, abattoirs, commercial factories, river water, community settings that directly discharge effluents to the river and river sediments. 

### 2.3. Sampling Strategy and Sample Sites

Based on the characteristics of the study area, the Msimbazi River basin was categorized into three segments. The upper part where the river originates is slightly populated, less prone to flooding with agriculture, irrigation and drinking points for livestock activities. The middle part with moderate to dense population, multiple activities such as agriculture and irrigation, animal farming, drinking points for livestock, abattoirs and commercial factories is prone to frequent flooding. The lower part is characterised by prolonged flooding, a high-density population, agricultural activities, fishing and discharge from community, industrial and the national hospital effluents, and it is the discharge point of the river. From the three categories, ten sampling sites were selected: two from the upper part, six from the middle and two from the lower part of the basin (Figure 1). Samples of river water, river sediments, crop soil and effluent from the community, factory, abattoir, hospital and veterinary settings that are directly discharged onto the river were collected. During sampling, information about the localities where samples were collected (sampling sites), dates of sampling, type of sample, sample origin, weight of the sample, transportation conditions, test parameters to be conducted, organism of interest, time of sample collection and analysis was recorded in a special form. 

### 2.4. Sample Collection and Processing

For each site, between three and seven samples were collected from different positions, making a total of 219 liquid and solid samples. The liquid samples were collected in a sterile 50 mL falcon tube (BD, Nairobi, Kenya). For river water, samples were collected from a distance of at least 0.5 m from the shore and at a depth of 20 and 50 cm. The community, industrial and hospital effluents were collected from the point where the final effluents entered into the river. Soil samples were collected at up to five different points per site using a sterile trowel and placed in a 50 mL sterile falcon tube (BD, Nairobi, Kenya). The tube was then placed in a zip-top bag. Liquid and solid samples were labelled according to the site of collection, placed in a cooler box containing ice packs and transferred to the laboratory for analysis within 3 h of collection. In each site the frequency of sampling was conducted once. The collection and processing of solid samples was conducted following the procedure described by [24], with some modifications. The liquid samples were processed as per [8], in which each individual sample was mixed with sterile 0.9% saline at a ratio of 1:1 and mixed to produce a homogenous solution. Approximately 1 g of the soil sample was made into suspension by adding 4 mL of normal saline and mixed thoroughly by vortexing. 

### 2.5. Isolation and Identification of Bacterial Strains

A loopful of sample suspension was inoculated onto MacConkey agar (Oxoid, Basingstoke, UK) without antibiotics and incubated at 37 °C aerobically for 24 h. A single colony from predominant morphologically similar colonies was picked from each plain MacConkey agar plate and subcultured in a Nutrient agar (Hi media, Mumbai, India). Colonies on the Nutrient agar were identified by colonial morphology, Gram stain, catalase and oxidase production [25] and various biochemical tests (Indole, Methyl red, Voges Proskauer, and Citrate utilisation tests) and were later confirmed by API 20E following the manufacturer’s recommendations (BioMérieux, Marcyl’Etoile, France) [26]. Briefly, a single colony was emulsified into sterile saline and filled in the compartments, then incubated at 37 °C for 18 to 24 h aerobically in a wet chamber of API 20 E strips. *Escherichia coli* and *Klebsiella pneumoniae* were identified to the species level.

### 2.6. Antimicrobial Susceptibility Testing

The antimicrobial susceptibility testing was done using the Kirby–Bauer disc diffusion method on Mueller Hinton agar (Becton, Dickinson and Company, New Jersey, USA) based on the Clinical Laboratory Standard Institute (CLSI) standards [27]. The antibiotics tested were ampicillin (10 µg), cefotaxime (30 µg), ceftriaxone (30 µg), nalidixic acid (30 µg), ciprofloxacin (5 µg), tetracycline (30 µg), chloramphenicol (30 µg), meropenem (10 µg) and imipenem (10 µg). One to two colonies from the pure culture of the identified lactose fermenters were emulsified into 5 mL of sterile saline. The suspension was adjusted to achieve turbidity equivalent to 0.5 McFarland standard solutions, emulsified using sterile cotton swabs onto Mueller Hinton agar plate and incubated at 37 °C for 16 to 18 h. The inhibition zone of each antimicrobial agent was measured using a ruler and the results were interpreted according to the CLSI standards 2019 [27]. *E. coli* strain ATCC 29522 and *K. pneumoniae* strain ATCC 700603 were used as controls. 

### 2.7. Screening and Confirmation of ESBL Production

Confirmed *E. coli* isolates were inoculated onto MacConkey agar containing 2 µg/mL cefotaxime for the preliminary screening of extended spectrum beta lactamase (ESBL) production [8]. Confirmation of ESBL production was performed using a combination disk diffusion method of cefotaxime (30 µg) alone and in combination with cefotaxime-clavulanic acid (10 µg) and ceftazidime (30 µg) alone and combination with ceftazidime-clavulanic acid (10 µg). *Klebsiella pneumoniae* (ATCC 700603) was used as a positive control (ESBL positive strain) and *E. coli* (ATCC 25922) was used as an ESBL negative strain; results were interpreted as per CLSI standards 2019 [27].

## 3. Results

### 3.1. Detection of Enterobacteriaceae Isolates and the Prevalence of Resistance from Effluent, River Water, River Sediment and Crop Soil

A total of 213 *Enterobacteriaceae* isolates were recovered from 219 samples (171, 27, 12 and 9 of effluents, river water, river sediments and crop soil, respectively). Approximately 60 samples yielded more than one isolate. Out of the recovered isolates, 45.5% (*n* = 97) were *Klebsiella pneumoniae* and 29.6% (*n* = 63) were *Escherichia coli*. Other *Enterobacteriaceae* (24.8%, *n* = 53) detected were *Serratia odorifera*, *Enterobacter aerogenes, Enterobacter cloacae*, *Klebsiella oxytoca*, *Pantoea* spp., *Citrobacter* spp., *Aeromonas hydrophila* and *Kluyvera* spp. (Table 1).

As shown in Figure 2, *K. pneumoniae* isolates were more resistant in effluent (27.9%) compared to the *E. coli* (26.6%) while the *E. coli* had a higher resistance in river water, sediment and crop soil than the *K. pneumoniae* (35 versus 25%), respectively. The *K. pneumoniae* isolates had higher resistance against nalidixic acid (54.6%) and ciprofloxacin (33.3%). The *E. coli* isolates were more resistant to ciprofloxacin (39.7%), ampicillin and trimethoprim/sulfamethoxazole (38%), respectively.

### 3.2. Prevalence of Antibiotic Resistance from Different Sample Locations

Overall, the level of resistance from the *K. pneumoniae* and *E. coli* isolates were higher in ampicillin (68.9%), nalidixic acid (60.4%), ciprofloxacin (36%) and trimethoprim/sulfamethoxazole (33.6%) throughout the sample locations. As shown in Figure 3, the level of antibiotic resistance increased from 28.3% in Kisarawe, where the river originates, to (59.9%) in Jangwani (the middle section) to 66.7% in Upanga West, where the river enters the Indian Ocean.

### 3.3. Multidrug Resistant E. coli and K. pneumoniae Isolates in Effluent, River Water, River Sediment and Crop Soil

Table 2 shows that out of 160 *E. coli* and *K. pneumoniae* isolates, 53.2% (*n* = 85) were multidrug resistant [28]. The most common resistance pattern determined was QNL/PEN/SUL (15 isolates), followed by QNL/PEN/TET (11 isolates) and CEP/QNL/PEN (nine isolates). Some of the *E. coli* and *K. pneumoniae* isolates were resistant to more than five classes of the antibiotic tested.

### 3.4. Prevalence of Quinolone Resistance, ESBL Producers and Carbapenem Resistant E. coli and K. pneumoniae from the Effluent, River Water, Sediment and Crop Soil

Out of 160 *E. coli* and *K. pneumoniae* isolates, 48.6% (*n* = 57) were found to be quinolone resistant [29], 14.4% (*n* = 23) were confirmed to be ESBL producers and 8.8% (*n* = 14) were carbapenem resistant. The ESBL producers were significantly resistant (*p* < 0.05) against trimethoprim/sulfamethoxazole and tetracycline and those that were quinolone resistant were significantly resistant to trimethoprim/sulfamethoxazole and tetracycline. The carbapenem resistant isolates had no significant resistance to the tested drugs except for tetracycline. The ESBL producers, quinolone resistant and carbapenem resistant isolates were more resistant to the tested antibiotics compared to the non-ESBL producers, non-quinolone resistant and non-carbapenem resistant isolates (Table 3). As shown in Table 4, both ESBL producers, quinolone resistant and carbapenem resistant isolates depicted various levels of resistance to the tested antibiotics and were significantly resistant compared to the isolates that were sensitive.

## 4. Discussion

This study was conducted in an ecosystem that is characterised by a high population, intensive agricultural and farming practices involving the use of manures, pesticides and antimicrobial agents, and is polluted with effluents and wastes from the largest pharmaceutical and commercial industries in the country [18,30,31]. The basin is located in Dar es Salaam, the largest city in Tanzania, with the highest population density of humans (3,133 humans/square kilometre) [32] and livestock in the country [33], and is the largest destination of livestock and livestock products from almost all areas of the country. In Africa, studies dealing with surveillance of the environment for AMR are very few [6,34] and to the best of our knowledge this is the first such study in Tanzania. In this study we used the recently described integrated systematic sampling approach [7] that ensures good quality data that can be shared and compared with data reported in other countries and contribute to the global picture. 

Overall, *K. pneumoniae* isolates were more resistant in effluent compared to the *E. coli* isolates which had higher resistance in river water, river sediment and crop soil. Both the *K. pneumoniae* and *E. coli* organisms are widely distributed in the community and wastewater settings and can easily acquire multiple resistance mechanisms as previously reported [26,34,35,36,37,38,39]. The levels of resistance to the tested antibiotics were higher in nalidixic acid (54.6%) and ciprofloxacin (39.7%) for *K. pneumoniae* and trimethoprim/sulfamethoxazole (38%) for *E. coli*, probably due to extensive use of these drugs in both animals and humans, uncontrolled disposal of drug leftovers and discharge of effluent into the environment and water bodies [23,40,41]. The levels of nalidixic acid, ciprofloxacin and trimethoprim/sulfamethoxazole found in this study are comparable with the one in Ethiopia and Tanzania, which reported that waste effluent discharged to the environment contains isolates of *E. coli* and *K. pneumoniae* that are resistant to multiple antibiotics [35,42]. 

In this study, more than half (53.2%) of the *E. coli* and *K. pneumonia* isolates from the effluent, river water, river sediment and crop soil exhibited multidrug resistance against three to seven classes of the antibiotic tested. The most frequent combination was observed in QNL/PEN/SUL, QNL/PEN/TET and CEP/QNL/PEN and might be attributed to extensive use in animals and humans and their release to the environment [41,43,44]. A previous study from Tanzania, Mozambique and Zambia reported quinolone, tetracycline, penicillin and sulfamethoxazole antibiotics were among the commonly used antimicrobial in animals and humans [14,45,46,47,48] with the consequence of increasing MDR organisms across the human, animal and environmental compartments. 

We found significantly lower levels of antibiotic resistance in Kisarawe and Pugu station, where the river originates, compared to the lower part of the basin (Jangwani and Upanga West areas), which discharges into the Indian Ocean. A possible explanation of these findings is that the basin is less contaminated at the upper part compared to the middle and lower basin due to anthropogenic activities and human settlement [31,49].

The overall prevalence of ESBL producers, quinolone and carbapenem resistance were 14.4, 48.6 and 8.8%, respectively, suggesting widespread use and release of these antimicrobial drugs and their active metabolites to the environment [50,51]. The acquisition of the gene encoding for resistance is through plasmids, transposons and integrons, which can spread very rapidly, thus posing a problem of treatment failure [34,50]. Our previous findings from the study area that compared genotypic and phenotypic results of MDR *E. coli* isolates, ESBL producers and quinolone resistance found that 80% of the isolates harboured *bla*CTX-M, 15% *aac*(*6*)-*lb*-*cr*, 10% *qnrB* and 5% *qepA*. None harboured TEM, SHV, *qnrA*, *qnrS*, *qnrC*, or *qnrD* [46,52]. The ESBL level found in this study was higher than that reported in soil and water samples in Tunisia and DRC Congo [36,41] and lower than previous findings from Tanzania and in Angola [8,53]. The level of quinolone in this study was lower (85%) than that reported from similar studies in Algeria and Togo [34,44]. These variations may be due to differences in relative use of the antibiotics between countries.

Our results, which show 27.9 and 35% of resistant *K.pneumoniae* and *E. coli* isolates, respectively, compare with other environmental surveillance studies that showed AMR levels between 33 and 100% in samples collected from rivers and streams, effluent and solid wastes [6]. Collectively, these results show a high level of environmental contamination with AMR and MDR bacteria in Africa, due to a high intensity of AMU in animals and humans, agriculture activities and weakness in regulation and disposal of antimicrobials in the environment [54]. 

We hypothesize that the sources of AMR bacteria in the basin are mainly the uncontrolled use of a wide range of antibiotics in humans and animals, use of manure and pesticides in agriculture, the release of effluents from different setups ranging from community, hospital and veterinary healthcare and commercial industries.

Therefore, we do recommend the following measures be instituted to address pollution of the environment with AMR bacteria in the Msimbazi basin: strengthen implementation of the available legislation, such as the national Environmental Management Act 2004 [55] to control for environmental contamination, the Animal Welfare Act 2008 [56] and the Veterinary Act 2003 [57] on proper animal husbandry and the Tanzania Food and Drugs Authority (TFDA) Act 2003 [58] on the proper handling and sale of both human and veterinary drugs; preventing the use of contaminated water for crop irrigation and establishment of multisectoral intervention and focusing on training and creating awareness of the magnitude and consequences of the AMR problem. We are also advocating for continuous monitoring of these measures to evaluate their impact on curbing environmental contamination with AMR organisms and antimicrobial agents. 

## 5. Conclusions

The level of AMR and MDR bacteria, including ESBL producers and quinolone resistant strains, seen in the Msimbazi basin is very high, which poses a risk to both human and animal health. Curbing AMR in the basin will require comprehensive and well-coordinated approaches that include continuous surveillance and stewardship on AMU and AMR in animals and humans; revising the approach on the implementation of regulatory bodies governing handling, distribution and sale of human and veterinary drugs; proper disposal of waste and effluents and improved agricultural practices along the river basin. Modelling will be required to assess the most effective set of approaches to embark on.

## Figures and Tables

**Figure 1 ijerph-18-08264-f001:**
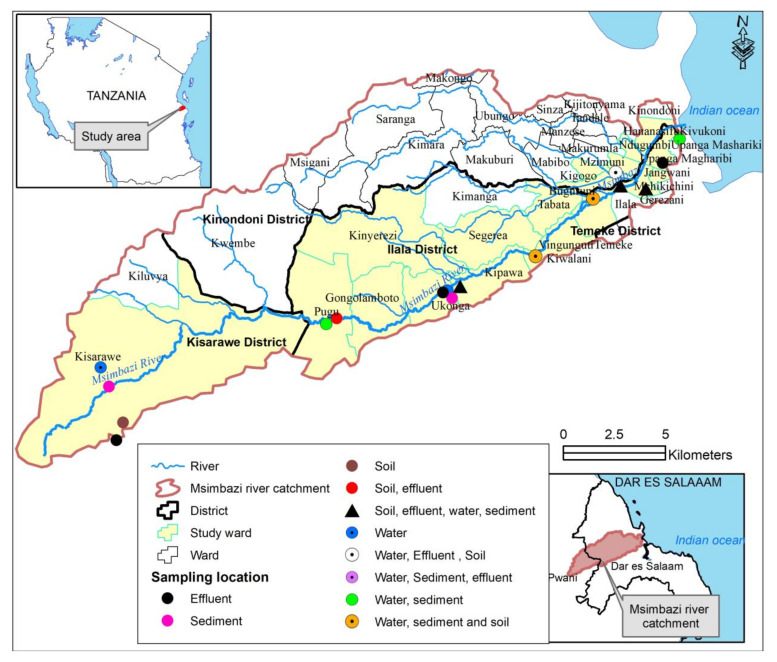
Map of Msimbazi river basin showing sample collection points and the sample type.

**Figure 2 ijerph-18-08264-f002:**
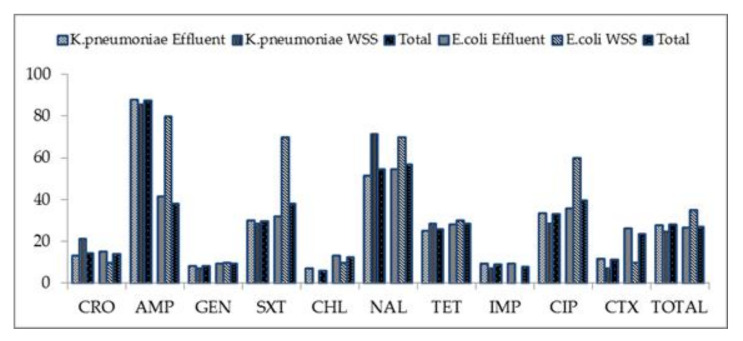
Percentage of antibiotic resistance from the effluent, water, sediments and soil isolates. **Key:** CRO, Ceftriaxone; AMP, ampicillin; GEN, gentamycin; SXT, trimethoprim/sulfamethoxazole; CHL, chloramphenicol; NAL, nalidixic acid; TET, tetracycline; IMP, imipenem; CIP, ciprofloxacin; CTX, cefotaxime; WSS, river water, sediment and crop soil.

**Figure 3 ijerph-18-08264-f003:**
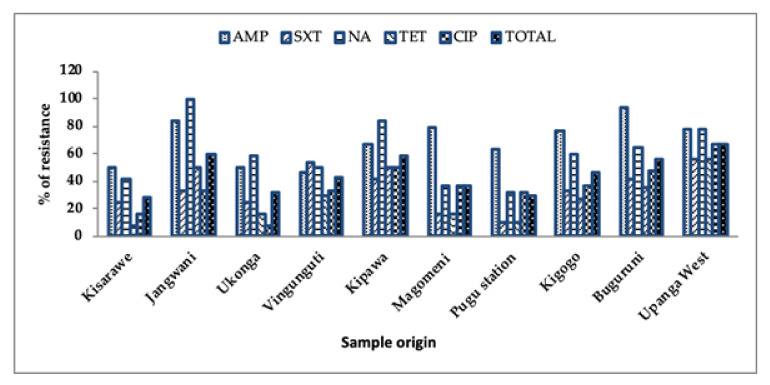
Resistance pattern of the antibiotics by samples’ geographical locations. Key: AMP, ampicillin; SXT, trimethoprim/sulfamethoxazole; NAL, nalidixic acid; TET, tetracycline; CIP, ciprofloxacin.

**Table 1 ijerph-18-08264-t001:** Distribution of bacterial species isolated from 219 effluent, water, sediment and soil samples.

Organism	Effluent (*n* = 176)	Water (*n* = 22)	Sediment (*n* = 10)	Soil (*n* = 5)	Total
	No (%)	No (%)	No (%)	No (%)	No (%)
*Klebsiella pneumoniae*	83 (85.6)	8 (8.3)	3 (3.1)	3 (3.1)	97 (100.0)
*Escherichia coli*	53 (84.1)	8 (12.7)	1 (1.6)	1 (1.6)	63 (100.0)
*Other organisms*	40 (75.5)	6 (11.3)	6 (11.3)	1 (1.9)	53 (100.0)

**Table 2 ijerph-18-08264-t002:** Overall resistance pattern of MDR isolates from community effluent, river water, river sediments and soil.

No of Antibiotics Classes	Resistant Pattern	No. of Isolates	Prevalence (%)
3	CEP/QNL/PEN	9	5.63
	SUL/PHE/PEN	3	1.88
	CEP/QNL/CAR	2	1.25
	PHE/TET/PEN	3	1.88
	QNL/PEN/SUL	15	9.34
	TET/CAR/QNL	3	1.88
	QNL/PEN/TET	11	6.88
	PEN/AMN/SUL	2	1.25
	CAR/CEP/PHE	3	1.88
	SUL/PHE/AMN	1	0.63
	CEP/QNL/AMN	2	1.25
4	PHE/TET/CAR/PEN	2	1.25
	PHE/TET/CAR/SUL	4	2.5
	CEP/QNL/PEN/TET	2	1.25
	CEP/QNL/PEN/CAR	3	1.88
	TET/CAR/CEP/PEN	1	0.65
	SUL/PHE/TET/QNL	3	1.88
	PEN/AMN/SUL/TET	3	1.88
5	SUL/PHE/TET/CAR/CEP	2	1.25
	TET/CAR/CEP/QNL/SUL	3	1.88
	CAR/CEP/QNL/PEN/SUL	1	0.65
6	CEP/QNL/PEN/AMN/SUL/TET	2	1.25
	SUL/PHE/TET/CAR/CEP/QNL	1	0.63
	TET/CAR/CEP/QNL/PEN/SUL	3	1.88
7	SUL/PHE/TET/CAR/CEP/QNL/PEN	1	0.63
Total		85	53.2

Key: QNL, quinolones; PHE, phenicols; AMN, aminoglycosides; PEN, penicillins; TET, tetracyclines; SUL, sulfonamides; CEP, cephalosporins; CAR, carbapenems.

**Table 3 ijerph-18-08264-t003:** Comparative distribution of quinolone resistant *E. coli* and *K.pneumoniae* isolated from different sample sources.

Antibiotic	% of Resistance *E. coli* and *K. pneumoniae* Isolates		*p*-Value	% of Resistance *E. coli* and *K. pneumoniae* Isolates		*p*-Value	% of Resistance *E. coli* and *K. pneumoniae* Isolates		*p*-Value
	ESBL Producers (*n* = 23)	Non-ESBL Producers (*n* = 137)		Quinolone Resistant (*n* = 57)	Non-Quinolone Resistant (*n* = 103)		Carbapenem Resistant (*n* = 14)	Non-Carbapenem Resistant (*n* = 146)	
AMP	NA	NA	NA	41 (71.9)	71 (68.9)	0.692	NA	NA	NA
GEN	14 (60.9)	12 (8.7)	0.992	14 (24.6)	9 (8.7)	0.994	13 (92.9)	13 (8.9)	0.824
SXT	17 (73.9)	41 (29.7)	0.036	31 (54.4)	22 (21.4)	0.000	8 (57.1)	47 (32.2)	0.419
CHL	12 (52.2)	11 (8.0)	0.432	13 (22.8)	8 (7.8)	0.555	12 (85.7)	12 (8.2)	0.444
TET	13 (56.5)	33 (24.1)	0.053	31 (54.4)	12 (11.7)	0.000	9 (64.3)	38 (26)	0.436

Key: AMP, ampicillin; GEN, gentamycin; SXT, trimethoprim/sulfamethoxazole; CHL, chloramphenicol; TET, tetracycline.

**Table 4 ijerph-18-08264-t004:** Comparison resistance levels between ESBL versus non-ESBL producers, quinolone resistant versus quinolone sensitive and carbapenem resistant versus carbapenem *E. coli* and *K.pneumoniae.*

Antibiotic	ESBL Producers (*n* = 23)		Quinolone Resistant (*n* = 57)		Carbapenem Resistant (*n* = 14)	
	R	S	R	S	R	S
AMP	NA	NA	41 (71.9)	16 (28.3)	14 (100)	0 (0.0)
GEN	14 (60.9)	9 (39.1)	14 (24.6)	43 (75.4)	13 (92.9)	1 (7.1)
SXT	17 (73.9)	6 (26.1)	31 (54.4)	26 (45.6)	8 (57.1)	6 (42.9)
CHL	12 (52.2)	11 (47.8)	13 (22.8)	44 (77.2)	12 (85.7)	2 (14.3)
TET	13 (56.5)	10 (43.5)	31 (54.4)	26 (45.6)	9 (64.3)	5 (35.7)

Key: AMP, ampicillin; GEN, gentamycin; SXT, trimethoprim/sulfamethoxazole; CHL, chloramphenicol; TET, tetracycline.

## Data Availability

The data generated are contained within the article.

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
