# Peer review of "Occurrence of Extended Spectrum Beta Lactamase (ESBL) Producers, Quinolone and Carbapenem Resistant Enterobacteriaceae Isolated from Environmental Samples along Msimbazi River Basin Ecosystem in Tanzania"

_ijerph, 2021, doi:10.3390/ijerph18168264_

Round 1

Reviewer 1 Report

The manuscript submitted by Kimera and coworkers for publication in IJERPH is soundness paper which describes the antimicrobial resistance of several bacterial strains in some certain region of Tanzania. Since the authors performed experiments about environmental AMR, this study is really appropriate for the standard of the journal and thus it can be accepted for publication.

The data are clearly reported, the AMR is highlighted by solid experiments and the overall merit of author is to be underlined the needed for intervention in this area with antimicrobial drugs campaign.

I have only some minor comments which regard the manuscript font which needs to be adjusted in Palatino Lynotype. It is very varied along the main text.

Please specify what AMU means before use the abbreviation and check similar mistakes in the text.

Please have a look to puntuaction and grammar mistakes in the main text.

Revise the references in [] in main text.

Author Response

S/N

COMMENTS

RESPONSE

REVIEWER 1

1

I have only some minor comments which regard the manuscript font which needs to be adjusted in Palatino Lynotype. It is very varied along the main text.

The entire manuscript has been changed to Palatino Linotype

2

Please specify what AMU means before use the abbreviation and check similar mistakes in the text.

The word “AMU” specified to read “antimicrobial use” when used for the first time

3

Please have a look to puntuaction and grammar mistakes in the main text.

The entire manuscript has been revised to look for punctuation and grammar mistakes

4

Revise the references in [] in main text.

The reference style has been revised to remove () and replace by [] throughout the manuscript

Reviewer 2 Report

Occurrence of Extended Spectrum Beta Lactamase (ESBL) Producers, Quinolone and Carbapenem Resistant Enterobacteriaceae Isolated From Environmental Samples Along Msimbazi river Basin Ecosystem in Tanzania by Kimera et. al. identifies the antimicrobial strains from the environmental samples. The study is comprehensive but lacking a few more datas before it could be accepted for publication. So, I recommend it for major revision.

 Abstract: “surveillance of AMR”: although mentioned in introduction, expands AMR when used first.

Introduction:

Change: “Due nature and complexity of the drivers” to

 “Due to nature and complexity of the drivers”

Methods: Author mentioned ref7 for sampling frame. Since it is critical part of the study, the authors could elaborate in detail the specified frame.

Many minor English corrections are required like missing articles, sentences construction errors that need to be fixed throughout the manuscript.

Major comments:

Result 3.1: The authors concluded the detection of enterobacteriaceae isolates and stated the presence of 45.5% (n=97) of Klebsiella pneumoniae and 26.9% (n=63) of Escherichia coli along with other strains like Serratia odorifera, Enterobacter aerogenes, Enterobacter cloacae, Klebsiella oxytoca, Pantoea spp, Citrobacter spp, Aeromonas hydrophila, and Kluyvera spp. However, author didn’t present the data for the same.

The authors have shown the antibiotic resistance data and statistics. Did the authors also check the antimicrobial resistance gene through genotyping? If not, can they support their findings or mention few references for the same??

Author Response

S/N

COMMENTS

RESPONSE

REVIEWER 2

Abstract: “surveillance of AMR”: although mentioned in introduction, expands AMR when used first.

The word “AMR” used for the first time in the abstract has been extended to read “antimicrobial resistance”

Introduction: Change: “Due nature and complexity of the drivers” to  “Due to nature and complexity of the drivers”

The sentence revised to read “Due to nature and complexity of the drivers”…

Methods:

Author mentioned ref7 for sampling frame. Since it is critical part of the study, the authors could elaborate in detail the specified frame.

Elaborations on the sampling frame selected for sampling has been made within the methodology section as requested

Many minor English corrections are required like missing articles, sentences construction errors that need to be fixed throughout the manuscript.

Revision has been made to correct for the English, missing articles and construction of the sentences

Major comments:

Result 3.1: The authors concluded the detection of enterobacteriaceae isolates and stated the presence of 45.5%(n=97) of Klebsiella pneumoniae and 26.9% (n=63) of Escherichia coli along with other strains like Serratia odorifera, Enterobacter aerogenes, Enterobacter cloacae, Klebsiella oxytoca, Pantoea spp, Citrobacter spp, Aeromonas hydrophila, and Kluyvera spp. However, author didn’t present the data for the same.

The data for other enterobacteriaceae detected has now been included which is 24.8% (n=53)

The authors have shown the antibiotic resistance data and statistics. Did the authors also check the antimicrobial resistance gene through genotyping? If not, can they support their findings or mention few references for the same??

We did not check the resistance gene through genotyping because in a previous analysis we did perform genotyping on some of these isolates to determine genes encoding for ESBL production and quinolone resistance.  We compared phenotypic with genotypic results of 20 MDR E. coli isolates, ESBL producers, and quinolone-resistant strains and found 80% harbored blaCTX-M, 15% aac(6)-lb-cr, 10% qnrB, and 5% qepA. None harbored TEM, SHV, qnrAqnrSqnrC, or qnrD

We have added a corresponding text in the discussion and provided references as requested.

Reviewer 3 Report

The authors described the occurrence and distribution of extended-spectrum β-lactamase (ESBL)-producing, carbapenem-resistant Enterobacteriaceae (CRE) and quinolone resistant Escherichia coli and Klebsiella spp. I recommend its publication. I have only one comment. Did you perform a statistical analysis? How many technical replicates did you have for each sample? Please provide more information.

Author Response

S/N

COMMENTS

RESPONSE

REVIEWER 3

The authors described the occurrence and distribution of extended-spectrum β-lactamase (ESBL)-producing, carbapenem-resistant Enterobacteriaceae (CRE) and quinolone resistant Escherichia coli and Klebsiella spp. I recommend its publication. I have only one comment. Did you perform a statistical analysis? How many technical replicates did you have for each sample? Please provide more information.

In this study we only collected one sample per site per site and performed statistical analyses as reported in the methodology. We did not have replicates.

Round 2

Reviewer 2 Report

Authors have addressed most of my concerns. The manuscript looks better now and can be considered for publication in given journal.